# Influence of Electrolyte Temperature on the Color Values of Black Plasma Electrolytic Oxidation Coatings on AZ31B Mg Alloy

**Aihua Yi [1], Zhongmiao Liao [1], Wen Zhu [1], Zhisheng Zhu [2], Wenfang Li [1,*], Kang Li [1], Ken Chen [1] and Shengkai Huang [1]**

[1] School of Materials Science and Engineering, Dongguan University of technology, Dongguan 523808, China; yiah@dgut.edu.cn (A.Y.); 2017141@dgut.edu.cn (Z.L.); zhuwen@dgut.edu.cn (W.Z.); 2017065@dgut.edu.cn (K.L.); chenk@dgut.edu.cn (K.C.); 2018005@dgut.edu.cn (S.H.)

[2] School of Materials Science and Engineering, South China University of Technology, Guangzhou 510640, China; zhuzs1@bngrp.com

[*] Correspondence: liwenfang@dgut.edu.cn; Tel.: +86-769-2286-2966

**Abstract:** A coating was prepared on an AZ31B Mg alloy substrate via black plasma electrolytic oxidation (PEO). The colorant $NH_4VO_3$ was added to $Na_2SiO_3$–$(NaPO_3)_6$ electrolyte at different temperatures (5, 15, 25, and 35 °C). The influences of electrolyte temperature on the structures, compositions, and color values of black PEO coatings were studied by UV–Visible, XRD, XPS, Raman, and SEM techniques. The results showed that the relative content of $V_2O_3$ and $V_2O_5$ was the key factor affecting the coating color value. At higher temperatures, more $NH_3$ escaped from the electrolyte and the $NH_3$ quantity participating in the reaction decreased, resulting in a decrease of $V_2O_3$ content, an increase in color value, and a darker coating. In the PEO process, $VO^{3-}$ mainly reacted to form $V_2O_5$, and then, the generated $V_2O_5$ reacted with $NH_3$ to form $V_2O_3$.

**Keywords:** plasma electrolytic oxidation; electrolyte temperature; color value

## 1. Introduction

Magnesium (Mg) alloys are the lightest metallic structural materials (~1.8 g/cm$^3$), and they possess outstanding properties, such as high specific strength, high specific stiffness, good castability, and high damping capacity [1,2]. Therefore, Mg alloys have many applications in 3C electronic products, and the transportation and aerospace industries. In particular, with the rapid development of artificial intelligence, high specific strength and good electromagnetic shielding are two of the most desirable mechanical properties in 3C electronic products. However, Mg and its alloys have low standard potential [3] ($E$ = −2.363 V), and they are electrochemically active, making them susceptible to corrosion in various environments. This poor corrosion resistance severely limits the widespread application of Mg alloys in many critical industrial fields. The surface coloring process has been used to prepare decorative coatings on Mg alloys for expanding their application space, such as in 3C products. Among the various surface techniques, plasma electrolytic oxidation (PEO), as a reliable and environment-friendly technique [4–6], has attracted attention for improving corrosion resistance and providing good adhesion between the coating and substrate.

After years of development, a composite electrolyte system of silicate [7], phosphate [8], aluminate [9], and their combination [10] has been established. Many researchers have invested much effort into examining the influences of electrical parameters such as density [11–13], voltage [14,15], duty cycle [16], frequency [17], and the electrolyte, including additive composition [1,18] and different concentration [19]. Electrolyte temperature, an important factor in PEO coating formation, is now

attracting increasing attention. Bosta et al. [20] have reported that at higher electrolyte temperatures, thinner and rougher coatings can be formed and that sillimanite and cristobalite phases increase significantly. Raj et al. [21] have found that electrolyte temperature has significant impacts on the growth and thicknesses of coatings. Zhai et al. [22] have reported that a higher electrolyte temperature improves the growth rate and increases coating thickness, but also reduces coating corrosion resistance. Bai et al. [23] studied the effects of positive pulse voltage on color value and found that the minimum coating color value was 23 when the positive pulse voltage was 440 V, with the copper solid solubility in the coating being vital to the color value. Li et al. [24] found that coatings prepared in a hot electrolyte contained more sillimanite and cristobalite phases than those prepared in a colder electrolyte and they believe that $VO_3^{-1}$ first loses electrons to generate $V_2O_3$ and then partially converts to $V_2O_5$. However, this does not take into account the effects of $NH_3$ concentration and high pressure in the microregion of the PEO reaction. According to the literature [24], vanadium oxide is the main reason for the coating being black. Thus, to date, few researchers have focused on the effects of electrolyte temperature on the color values of black PEO coatings, and the mechanisms of electrolyte temperature's effects on color value remain unclear too.

In this study, a black PEO coating was prepared on an AZ31B Mg alloy surface by adding $NH_4VO_3$ into $(NaPO_3)_6$–$Na_2SiO_3$ electrolyte at different temperatures (5, 15, 25, and 35 °C). The structure, composition, morphology, roughness, and color value of each of the PEO coatings were then characterized. The high temperature and pressure in the reaction microregion and the presence of $NH_3$ in the electrolyte were considered, and based on chemical reaction thermodynamics, the reason for the influence of electrolyte temperature on the color value of black PEO coating was analyzed.

## 2. Materials and Methods

AZ91B Mg alloy was used in this study; the alloy's composition is shown in Table 1; and the sample's size was $40 \times 40 \times 3$ mm$^3$. The samples were ground successively using abrasive paper, with grit 200 #, 400 #, 600 #, 800 #, and finally 1200 #. Then, they were rinsed with deionized water and cleaned with ethanol. The PEO coatings were created using a 60 kW bipolar pulse power supply, equipped with a cooling system, which held a different constant temperature, recorded the electrical parameters with a data logging system, and had a stainless steel sheet serve as the cathode. The constant current mode was selected for the power supply and the electrical parameters, and the composition of the electrolyte is shown in Tables 2 and 3, respectively. Four different electrolyte temperatures (5, 15, 25, and 35 °C) were examined in this study. The $Na_2SiO_3$, $(NaPO_3)_6$, $NH_4VO_3$, KF, Na-Citrate, NaOH, and EDTA were successively added to 1 L of distilled water, which was mechanically stirred to dissolve and set aside. Then we put the prepared electrolyte solution into the circulating water (the electrolyte temperature was controlled through the circulating water of the electrolyte outside tank) and set the electrical parameters according to Table 2. We turned on the power and started the reaction after the electrolyte temperature was the same as that of the circulating water. The maximum voltage during the reaction was about 240 V.

**Table 1.** Composition of AZ31 Mg alloy.

| Element | Mg | Al | Zn | Ca | Si | Mn | Fe | Cu | Ni |
|---------|-----|---------|---------|------|------|---------|-------|------|-------|
| Content/% | Bal. | 2.5–3.5 | 0.6–1.4 | 0.04 | 0.08 | 0.2–1.0 | 0.003 | 0.01 | 0.001 |

**Table 2.** Electrical parameters.

| Current Density/A·dm$^2$ | Frequency/Hz | Duty Cycle/% | Time/min |
|---------------------------|--------------|--------------|----------|
| 4 | 800 | 35 | 10 |

**Table 3.** Composition of electrolyte.

| Chemical Component | $Na_2SiO_3$ | $(NaPO_3)_6$ | $NH_4VO_3$ | KF | Na-Citrate | NaOH | EDTA |
|---|---|---|---|---|---|---|---|
| Content g/L | 15 | 20 | 10 | 25 | 3 | 3 | 5 |

*Coatings Characterization*

A Lambda 950 UV/VIS/NIR spectrophotometer (PerkinElmer, Inc., Waltham, MA, USA) with a D65 illuminant was applied to measure the *L\**, *a\**, and *b\** of the black PEO coating. The color value of the black PEO coating was characterized by the CIE 1976 (*L\**, *a\**, *b\**) chromaticity system, with *L\**, *a\**, and *b\** representing the degree of lightness, and we covered a wide range from black (0) to white color (100). The color value is governed by |*L\**| + |*a\**| + |*b\**| and a smaller value suggests deeper color. Every panel of samples was measured at three different points to obtain reliable results by averaging.

Scanning electron microscopy (SEM) was performed with an acceleration voltage of 20 kV on a Nova Nanosem 430 (FEI Co., Hillsboro, OR, USA), which was equipped with a Quantax Silicon Drift Detector for energy dispersive X-ray spectroscopy (EDS).

The X-ray photoelectron spectroscopy (XPS) measurement was carried out using a Kratos Axis Ultra DLD (Kratos Analytical, Ltd., Manchester, UK) equipped with a standard Al Kα X-ray source (1486.6 eV) and a hemispherical analyzer. The pressure in the specimen chamber was held at ~$1 \times 10^{-9}$ mbar and the binding energy scale was adjusted using the C 1*s* line fixed at 284.6 eV.

The phase composition of coating was analyzed by X-ray diffraction (XRD) (Malvern Panalytical B.V., Almelo, The Netherlands) with a constant glancing angle of 0.5° and a scan range of 10°–90°. The PEO coating was scraped from the surface of the substrate and then analyzed.

The oxidation of the coating was analyzed using a micro Raman spectrometer (LabRAM Aramis, Horiba Jobin Yvon Ltd., Kyoto, Japan) with 532 nm wavelength and a scan range of 0–2000 cm$^{-1}$.

The coating thickness was determined using a thickness gauge (Phynix GmbH & Co. KG, Neuss, Germany), with a measurement deviation of ±1%, and 10 data points were obtained on both sides of each sample.

## 3. Results

*3.1. Color Values and PEO Coating Thickness*

The color value and optical photos of coatings produced at different electrolyte temperatures (Table 4 and Figure 1, respectively) showed that, with increasing temperature, the coating color gradually became shallower, and when the temperature was no more than 25 °C, the coating color difference was so small that it was almost indistinguishable to the naked eye. However, when the electrolyte temperature was up to 35 °C, the color changed greatly, the result being brown. The coating color values were not much different when the temperature was not higher than 25 °C, with values of ~25 (Table 4). However, when the temperature was above 25 °C, the color value increased from 25.82 at 25 °C to 29.03 at 35 °C, which indicated that the higher the temperature was, the lighter the coating color. The coating thickness at different temperatures showed that the higher the temperature was, the thinner the coating (Table 5).

**Table 4.** The color value of the coating.

| T/°C | 5 | 15 | 25 | 35 |
|---|---|---|---|---|
| The color value | 24.78 | 24.87 | 25.82 | 29.03 |

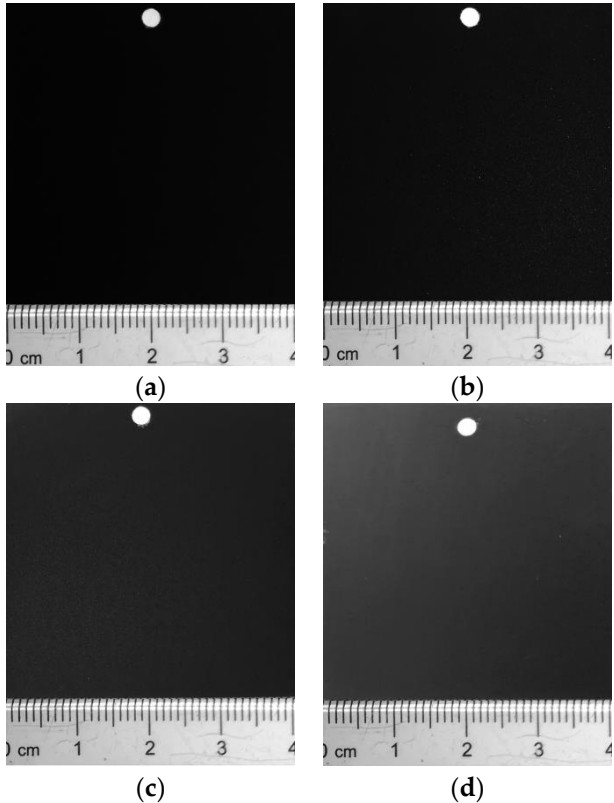

**Figure 1.** The optical photographs of the samples fabricated at different electrolyte temperatures: (**a**) 5 °C; (**b**) 15 °C; (**c**) 25 °C; (**d**) 35 °C.

**Table 5.** Thickness of the coating.

| T/°C | 5 | 15 | 25 | 35 |
|---|---|---|---|---|
| Thickness/μm | 14.54 | 12.39 | 8.97 | 7.54 |

## 3.2. XRD Analysis

The XRD patterns of PEO coatings formed at different electrolyte temperatures showed that electrolyte temperature had little effect on phase composition (Figure 2). The main coating components obtained at four different temperatures were $V_2O_3$, $Mg_2SiO_4$, $V_2O_5$, $Al_2O_3$, MgO, and $MgAl_2O_4$, and only the relative contents were different. With increasing temperature, the peak height and area of $V_2O_3$ gradually decreased, indicating that the $V_2O_3$ content decreased. At the same time, Mg and $Al_2O_3$ peaks began to appear in the coating when the electrolyte temperature was 25 °C. This might be explained by the fact that, when the coating thickness decreased as electrolyte temperature was increasing, Mg in the substrate was easily detected by XRD. Meanwhile, on the coating thickness decrease, Al can easily migrate from the substrate to the coating and form oxides. In addition, the peak related to MgO phase shifted to the lower angles, as compared to the standard peak position of MgO phase. This was because, at a high temperature, Mg in the substrate underwent arc breakdown and combined with O to form MgO, after which it flowed to the substrate surface and was rapidly condensed by the electrolyte, resulting in stress in the crystalline MgO phase and related XRD peak offsets. In addition, $Mg_2SiO_4$ phase was found in each coating, which was consistent with the literature [25]. In addition, at high temperatures, MgO will further form $Mg_2SiO_4$ and $MgAl_2O_4$ phases with $SiO_3^{2-}$ and $Al_2O_3$.

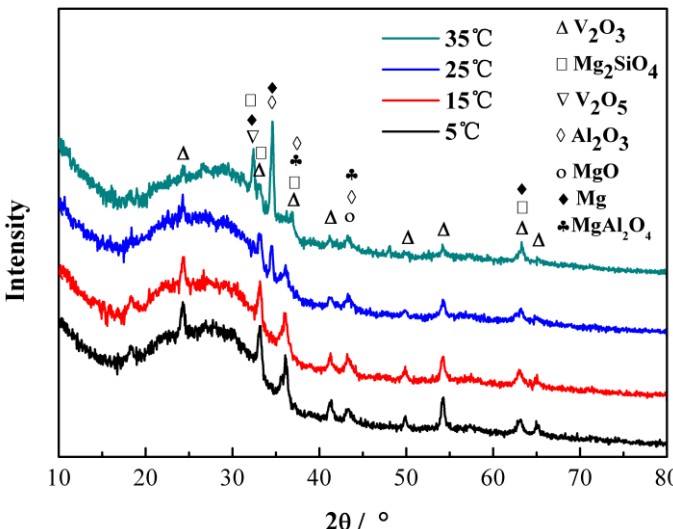

**Figure 2.** XRD spectra of PEO coatings fabricated at different electrolyte temperatures.

The XRD patterns showed that there were no $MgF_2$ and $Mg_3(PO_4)_2$ phases in the coating, but there were large amounts of $(NaPO_3)_6$ and $F^-$ in the electrolyte. The reason for this was that $F^-$ easily formed $MgF_2$ with Mg [1] and the layer was deposited on the substrate surface. As the reaction progressed, the formed coating hindered $F^-$ migration to the substrate, such that $MgF_2$ mainly existed near the substrate surface under the coating. As the grazing-incidence mode was adopted in this study, the test depths were small, such that $MgF_2$ was detected. Although the coating contained magnesium phosphate, in accordance with the literature [26], P in the membrane layer mainly existed in an amorphous form, without a diffraction peak.

### 3.3. Raman Analysis

The Raman spectra of coatings prepared at different temperatures showed that the absorption around 213 and 239 cm$^{-1}$ could be assigned to $V_2O_3$ [27] and peaks around 880 and 992 cm$^{-1}$ to $V_2O_5$ [28] (Figure 3). All coatings, obtained at different temperatures, contained $V_2O_5$. With temperature increasing, the $V_2O_3$ absorption peak height gradually weakened, and at 35 °C, the peak significantly decreased, indicating that increased temperature did not stimulate $V_2O_3$ generation.

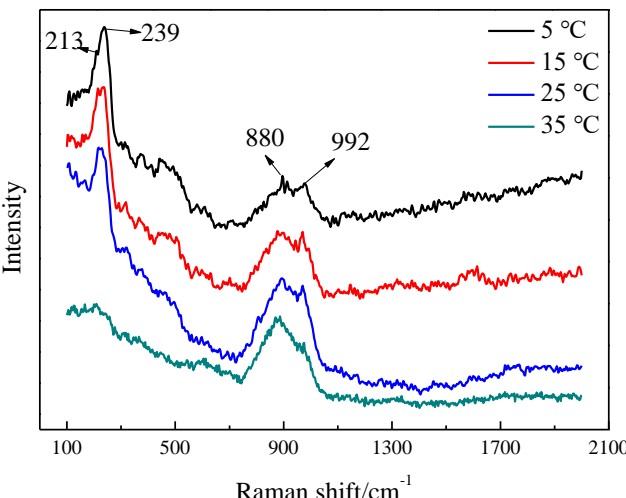

**Figure 3.** The Raman spectrum of PEO coatings with different electrolyte temperatures.

### 3.4. XPS Analysis

Examination of V-$2p_{3/2}$ peaks in the coating produced under different temperatures showed that the core-level spectra could be fitted by two components with binding energies at ~515.3 and 571.6 eV, which were assigned to $V_2O_3$ and $V_2O_5$ [29], respectively (Figure 4). The peak area in the XPS spectra reflected the content of an element to some extent, such that the peak area can be used to compare the relative contents of different valence states of the same element. The peak areas of $V^{3+}$ and $V^{5+}$ components in coatings obtained at different temperatures were integrated, and was $R_{VO}$ used to represent the relative contents of $V_2O_3$ and $V_2O_5$ ($R_{VO} = V^{3+}/V^{5+}$). The $R_{VO}$ results showed that, with the increasing temperature, $R_{VO}$ decreased from 2.93 at 5 °C to 0.30 at 35 °C, indicating that the $V_2O_3$ content decreased significantly, and at 35 °C, vanadium oxides mainly existed in the form of $V_2O_5$ (Table 6). These results are consistent with the results of Raman analysis.

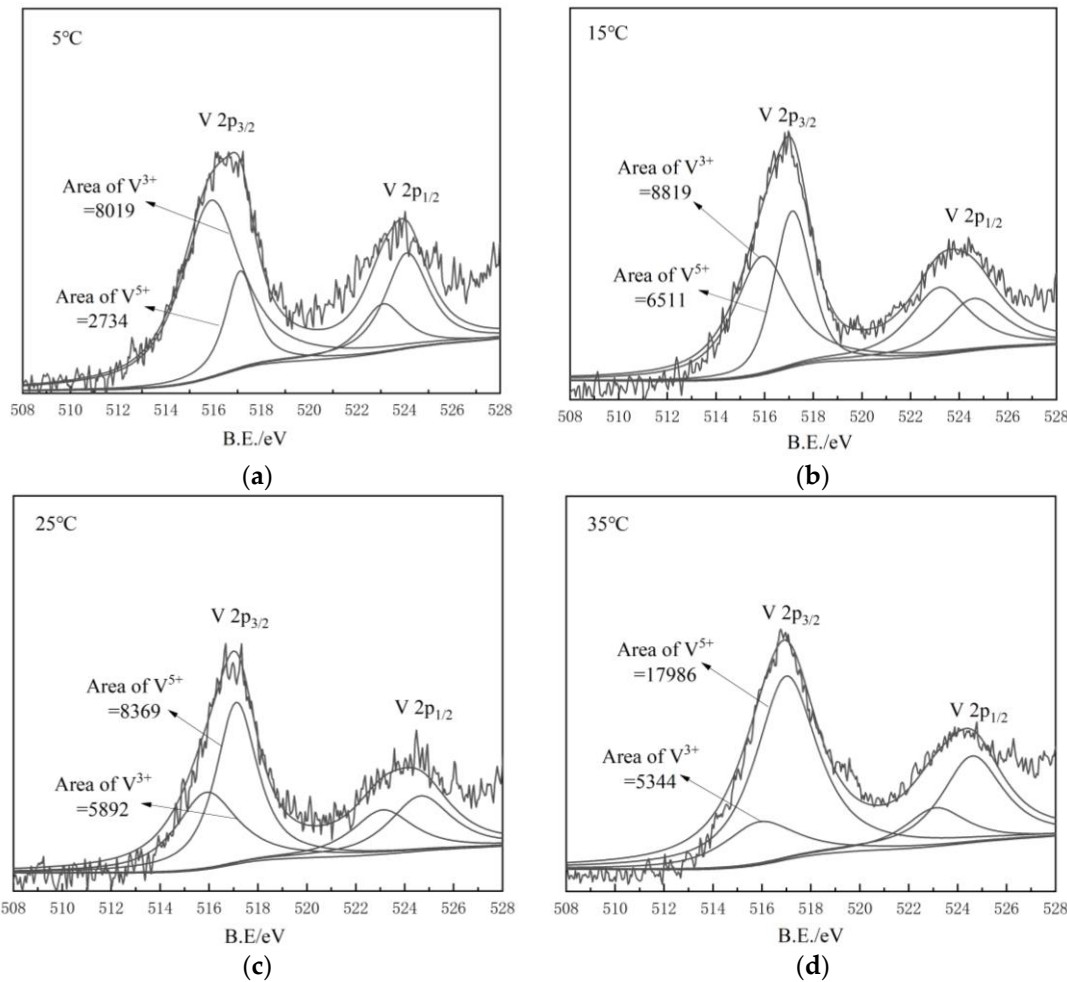

**Figure 4.** V $2p3/2$ lines in PEO coatings fabricated at different electrolyte temperatures: (**a**) 5 °C; (**b**) 15 °C; (**c**) 25 °C; (**d**) 35 °C.

**Table 6.** $R_{VO}$ values.

| T/°C | 5 | 15 | 25 | 35 |
|---|---|---|---|---|
| $R_{VO}$ | 2.93 | 1.35 | 0.70 | 0.30 |

### 3.5. SEM Analysis

The surface morphologies of coatings produced at different temperatures and viewed at different magnifications are shown in the Figure 5. From the figure, with increasing temperature, hole sizes on coating surfaces first decreased and then increased. When the temperature rose to 35 °C, the coating surface was bulging with large holes. The number of holes increased significantly with the temperature increase and holes inside the coating appeared to be interconnected.

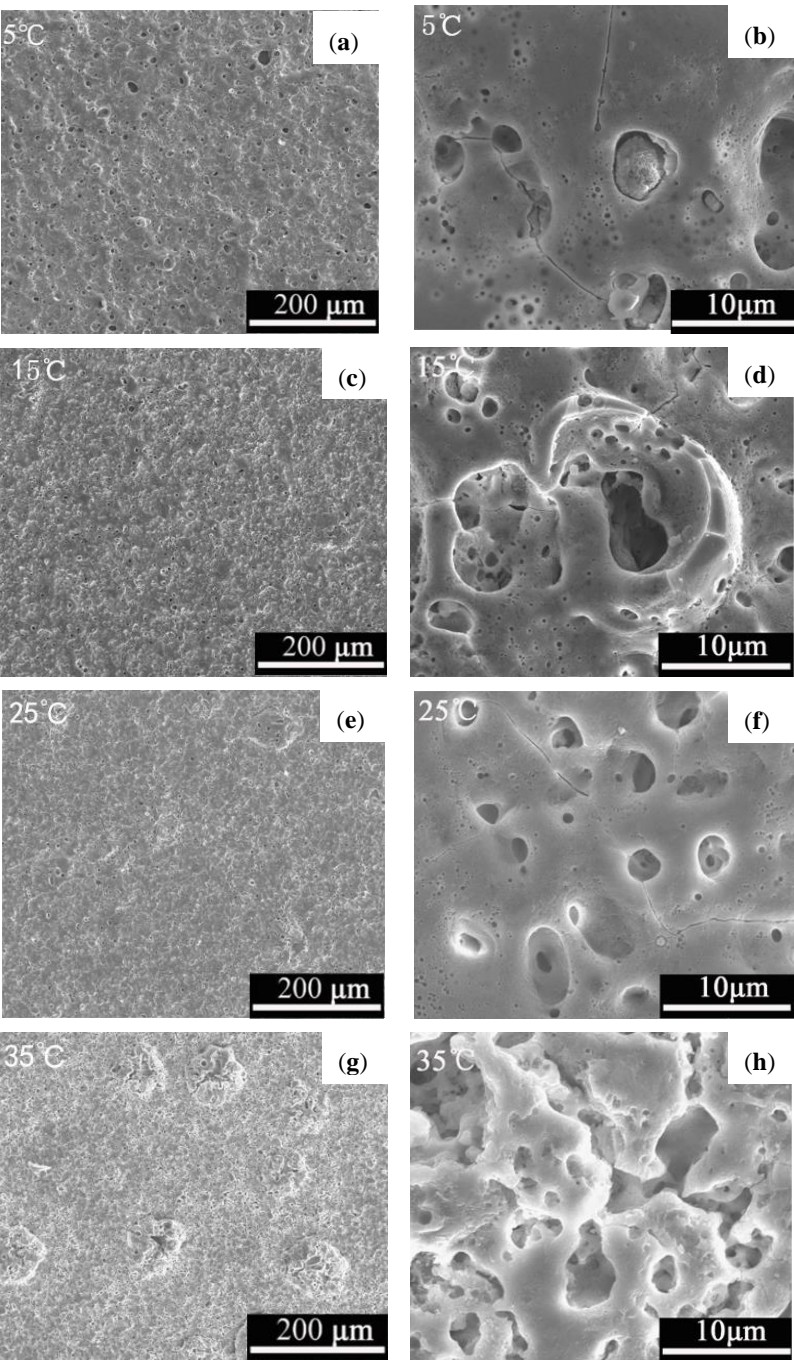

**Figure 5.** Morphologies of PEO coatings prepared at different electrolyte temperatures (500×: (**a**,**c**,**e**,**g**); 10,000×: (**b**,**d**,**f**,**h**)).

The examination of coating cross-sections showed that coating thickness gradually decreased with electrolyte temperature increase, and it is consistent with the coating thickness test results

(Figure 6 and Table 5). There were discharge channels and discharge holes in coatings prepared at different temperatures. As temperature increased, the discharge channels and holes gradually increased and coatings became "loose." Therefore, as electrolyte temperature rose, the coating interior density decreased.

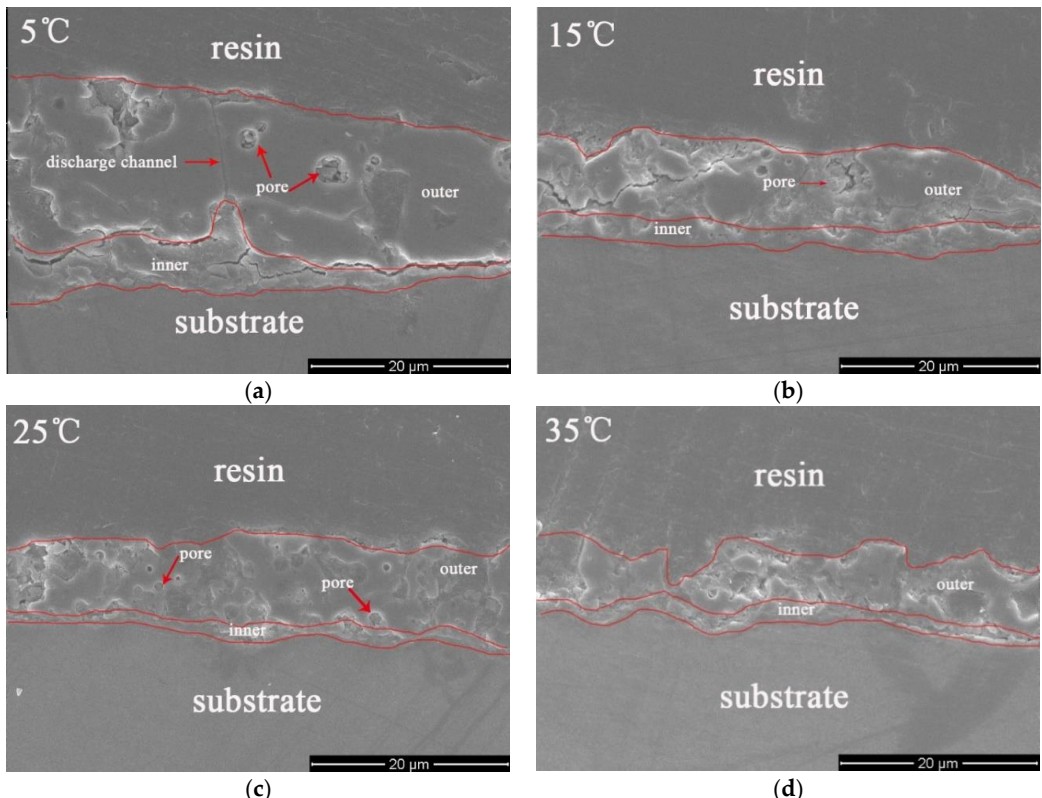

**Figure 6.** Cross-section SEM images of the PEO coatings prepared at different electrolyte temperatures: (**a**) 5 °C; (**b**) 15 °C; (**c**) 25 °C; (**d**) 35 °C.

The above analysis was confirmed by analyzing a coating (15 °C) section (15 °C) by SEM line scanning (Figure 7), which allowed mapping of the main element distributions (Figure 8). Fluorine was mainly distributed at the coating/substrate junction, while P element was evenly distributed in the whole coating. At the same time, it was found that Al content gradually decreased with increasing coating thickness, which was because Al came from the substrate and gradually migrated to the coating's surface under the arc breakdown action. The thicker the coating was, the lower the Al content.

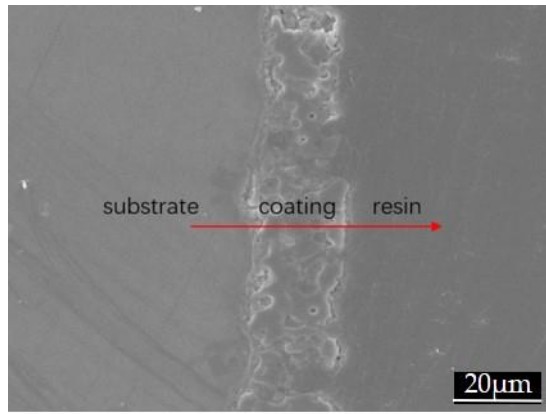

**Figure 7.** Cross-sectional SEM morphology of the PEO coating.

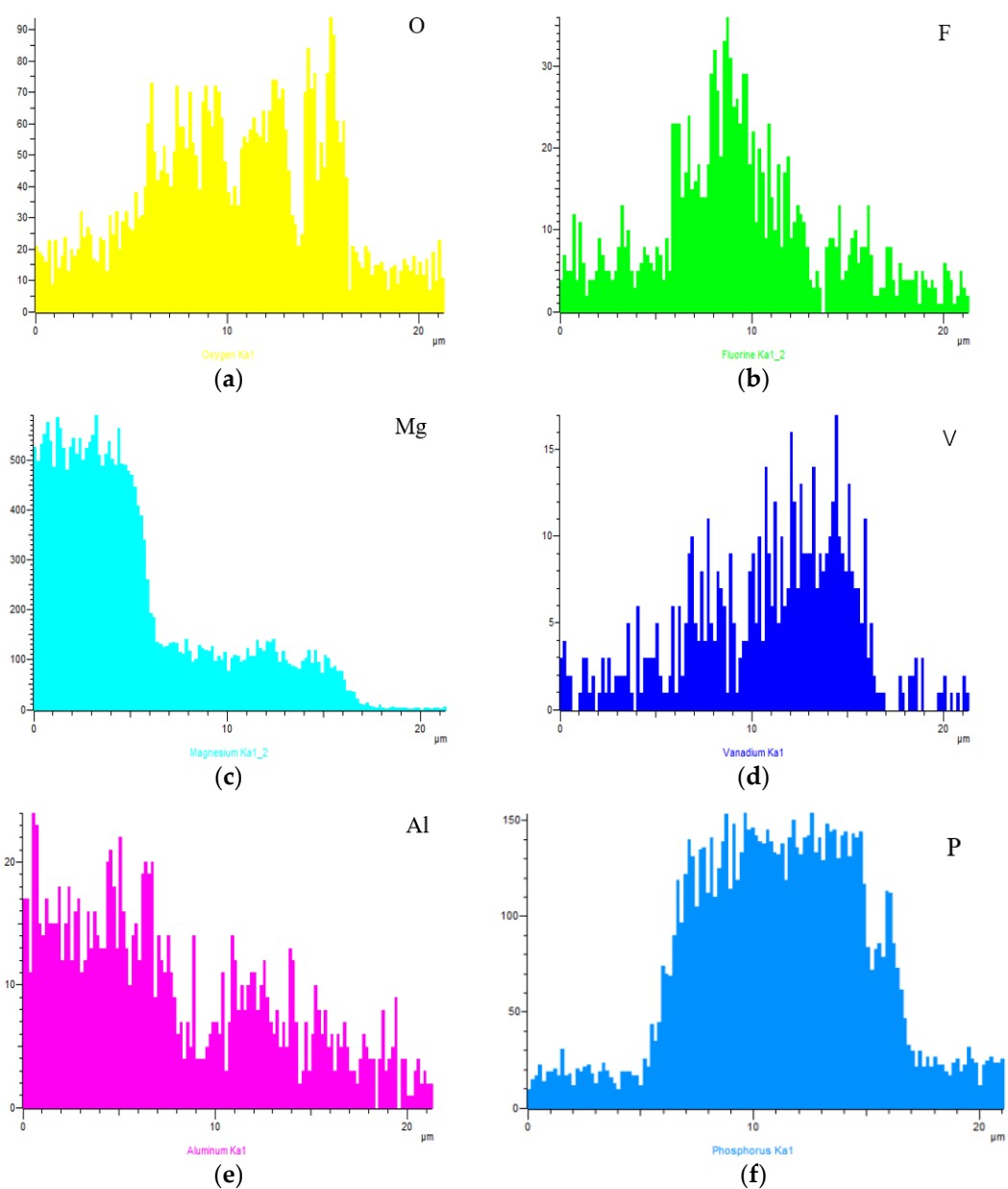

**Figure 8.** Element distributions of (**a**) O, (**b**) F, (**c**) Mg, (**d**) V, (**e**) Al and (**f**) P by line scanning along the arrow shown in Figure 7.

## 4. Discussion

*The Mechanism of the Color Value Change of the Black PEO Coating*

From the above analysis, V compounds observed in coatings were mainly $V_2O_3$ and $V_2O_5$, and as the temperature increased, $V_2O_3$ content decreased and $V_2O_5$ increased, which was also the key to the observed coating color value variation. As is well known, $V_2O_3$ is black and $V_2O_5$ is brown. Respectively, when $V_2O_3$ content in the coating increases, the color value becomes lower and coating color becomes darker. According to the relevant literature [24,30], the possible reactions of vanadium in the PEO process are as follows:

$$4VO_3^- - 4e^- \rightarrow 2V_2O_5 + O_2\uparrow \tag{1}$$

$$4VO_3^- - 4e^- \rightarrow 2V_2O_3 + 3O_2\uparrow \tag{2}$$

$$V_2O_5 \rightarrow V_2O_3 + O_2\uparrow \tag{3}$$

$$3V_2O_5 + 4NH_3 \rightarrow 3V_2O_3 + 6H_2O + 2N_2\uparrow \tag{4}$$

whether all or part of the above four reactions occur requires further analysis. We can analyze them from the perspective of thermodynamics. According to Gibbs free energy, $\Delta G = \Delta H - T\Delta S$, for these reactions to occur spontaneously, the condition of $\Delta G$ is $< 0$ must be satisfied. The relevant thermodynamic data, obtained from Lange's Handbook of Chemistry, were substituted into the equation (Table 7) and the results are shown in Table 8. Reaction (1) can be spontaneous if $T$ is $> 1698$ K, reaction (2) at $T > 1827$ K, and reaction (3) at $T > 1928$ K. For reaction (4), with $\Delta H$ at $< 0$ and $\Delta S > 0$, $\Delta G < 0$ at any temperature, which means that reaction (4) is always spontaneous. Judging from the spontaneous reaction temperature alone, the sequence of these four reactions should be (4) > (1) > (2) > (3), but the prerequisite for (4) to occur is the existence of $V_2O_5$. For that reason, reaction (4) must be after (1).

**Table 7.** Thermodynamic data.

| Thermodynamic Data | $VO_3^-$ (aq) | $V_2O_5$ (s) | $V_2O_3$ (s) | $O_2$ (g) | $N_2$ (g) | $H_2O$ (l) | $NH_3$ (aq) |
|---|---|---|---|---|---|---|---|
| $\Delta_f H_m^\circ$ (kJ·mol$^{-1}$) | −888.3 | −1550.59 | −1218.3 | 0 | 0 | −285.83 | −80.29 |
| $S_m^\circ$ (J·mol$^{-1}$·K$^{-1}$) | 50.2 | 130 | 98.3 | 205.152 | 191.5 | 69.91 | 111.29 |
| $\Delta_f G_m^\circ$ (kJ·mol$^{-1}$) | −783.7 | −1419.63 | −1139.3 | 0 | 0 | −237.18 | −26.57 |

**Table 8.** The minimum temperature of spontaneous reaction.

| Reactions | Standard State (T/K) | Nonstandard State (T/K) (Considering Gas Partial Pressure) |
|---|---|---|
| (1) $4VO_3^- - 4e \rightarrow 2V_2O_5 + O_2\uparrow$ | 1698 | 2183 |
| (2) $4VO_3^- - 4e \rightarrow 2V_2O_3 + 3O_2\uparrow$ | 1827 | 2541 |
| (3) $V_2O_5 \rightarrow V_2O_3 + O_2\uparrow$ | 1928 | 2260 |
| (4) $V_2O_5 + 4NH_3 \rightarrow 3V_2O_3 + 6H_2O + 2N_2\uparrow$ | - | - |

However, the above results only addressed the minimum temperature at which the reaction occurred spontaneously in the standard state, without considering the high temperature and pressure in the reaction microregion during the PEO process. The literature shows that the instantaneous pressure in the reaction microregion can reach $1 \times 10^8$ Pa or even higher [31], and the temperature can reach 6000 K or even higher [1,31–33]. If gas pressure is taken into account, what is the lowest temperature at which the above reaction spontaneously proceeds? Assuming that $P_{gas} = 10^8$ Pa, according to the Van't Hoff equation, the $\Delta G$ of Equations (5)–(7) are:

$$\Delta_r G_1 = 2\Delta_r G_{V2O5}(T) + \Delta_r G_{O2}(T) - \Delta_r G_{VO3}^-(T) + RT\ln(P_{O2}/P^\theta) \tag{5}$$

$$\Delta_r G_2 = 2\Delta_r G_{V2O3}(T) + 3\Delta_r G_{O2}(T) - \Delta_r G_{VO3}^-(T) + RT\ln(P_{O2}/P^\theta)^3 \tag{6}$$

$$\Delta_r G_3 = \Delta_r G_{V2O3}(T) + \Delta_r G_{O2}(T) - \Delta_r G_{V2O5}(T) + RT\ln(P_{O2}/P^\theta) \tag{7}$$

Additionally,

$$\Delta_r G_{V2O5}(T) = \Delta_r H_{V2O5}(T) - T\Delta_r S_{V2O5}(T) \approx \Delta_r H_{V2O5}(298.15\ K) - T\Delta_r S_{V2O5}(298.15\ K) \tag{8}$$

$$\Delta_r G_{V2O3}(T) = \Delta_r H_{V2O3}(T) - T\Delta_r S_{V2O3}(T) \approx \Delta_r H_{V2O3}(298.15\ K) - T\Delta_r S_{V2O3}(298.15\ K) \tag{9}$$

$$\Delta_r G_{O2}(T) = \Delta_r H_{O2}(T) - T\Delta_r S_{O2}(T) \approx \Delta_r H_{O2}(298.15\ K) - T\Delta_r S_{O2}(298.15\ K) \tag{10}$$

$$\Delta_r G_{VO3}^-(T) = \Delta_r H_{VO3}^-(T) - T\Delta_r S_{VO3}^-(T) \approx \Delta_r H_{VO3}^-(298.15\ K) - T\Delta_r S_{VO3}^-(298.15\ K) \tag{11}$$

If the reaction is spontaneous, then $\Delta G$ is < 0 and the data from Table 7 can be substituted into Equations (5)–(7), with the minimum temperatures of Equations (5)–(7) at 2183, 2541, and 2260 K, respectively (Table 8). As the instantaneous temperatures in the reaction microregion can reach 6000 K or even higher, the above reactions could occur under the conditions of PEO formation.

According to the present results, for $V_2O_5$, reactions (3) and (4) were competitive reactions, but since reaction (4) was always spontaneous, reaction (3) was almost impossible, such that $V_2O_5$ mainly reacted with $NH_3$ to generate $V_2O_3$ (Table 8). For reactions (1) and (2), as the spontaneous reaction temperature of reaction (1) was significantly lower than that of reaction (2) in the nonstandard state, $V_2O_5$ was generated by the main reaction of $VO^{3-}$ in the PEO process, and then, the generated $V_2O_5$ reacted with $NH_3$ to generate $V_2O_3$.

To verify whether the presence of $NH_3$ promoted $V_2O_3$ generation, $NH_4VO_3$ was replaced with $NaVO_3$ that was equimolar to the normal $VO^{3-}$ and the pH values were adjusted to be the same in both cases. The results showed that, when the temperature was 25 °C, the coating color value was as high as 36.03 when $NaVO_3$ was added, while that with $NH_4VO_3$ was 25, indicating that reaction (4) had occurred and that $NH_3$ played a key role in the formation of black PEO coating. Therefore, in this study, the main reactions of *V* in the PEO were (1) and (4).

Thus, the temperature influences coating color value. With increased temperature, $NH_3$ escape increased in the electrolyte, leading to decreased $NH_3$ in the reaction, such that the amounts of $V_2O_3$ decreased and $V_2O_5$ increased, yielding an $R_{VO}$ decrease. Finally, the coating mainly presented the brown color of $V_2O_5$.

## 5. Conclusions

- The $V_2O_3/V_2O_5$ content ratio is the main reason for coating color change with the variation of electrolyte temperature. At higher temperatures, the $V_2O_5$ content is higher.
- With temperature increase, $R_{VO}$ decreased from 2.93 at 5 °C to 0.30 at 35 °C, with the coating color value ranging from 24.78 at 5 °C to 29.03 at 35 °C.
- In the process of PEO formation, $V_2O_5$ was mainly generated by $VO^{3-}$ losing electrons and then $V_2O_5$ reacting with $NH_3$ in solution to form $V_2O_3$. The main reactions were $4VO_3^- - 4e^- \rightarrow 2V_2O_5 + O_2$ and $3V_2O_5 + 4NH_3 \rightarrow 3V_2O_3 + 6H_2O + 2N_2\uparrow$.
- The electrolyte temperature changed the coating microstructure. As temperature increased, the coating hole diameters decreased and hole numbers increased, with the coating thickness gradually decreasing.

**Author Contributions:** Software: Z.L., Z.Z., and W.Z.; methodology: K.C. and S.H.; validation: K.L., K.C., and S.H.; data analysis, W.L. and A.Y.; writing—original draft preparation, A.Y., Z.Z., and W.Z.; writing—review & editing, W.L. and A.Y.; data curation: W.Z. and Z.Z.; supervision: W.L. All authors have read and agreed to the published version of the manuscript.

**Funding:** This research has been supported by the Dong Guan Innovative Research Team Program Research, Start-up funds of DGUT (GC300501-087 and GC300502-045), the Natural Science Foundation of Guangdong Province (grant number 2018A030310024), the Guangdong Basic and Applied Basic Research Foundation (grant numbers 2019A1515110466 and 2019A1515110913), the Development Project (Key) of Dongguan Social Science and Technology (grant number 2020507140151), the Guangdong Research Center of High Performance Light Alloys Forming Technology, Novel Light Alloy, and the Process Technology Key Laboratory of Dongguan City.

**Conflicts of Interest:** The authors declare there is no conflict of interest regarding the publication of this paper.

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
