# Peer review of "Influence of Electrolyte Temperature on the Color Values of Black Plasma Electrolytic Oxidation Coatings on AZ31B Mg Alloy"

_coatings, doi:10.3390/coatings10090890_

Round 1

Reviewer 1 Report

Authors presented very interesting development of black PEO coatings on magnesium alloy. Considering vanadium presence the coating can be possibly used in fields of photocatalysis, catalysis. Presented process seems to be easily applicable in industry, however further research is required.

  1. Please correct the editorial part of the work (No spaces before citation, in line 64 space missing before unit ‘mm’, figure 1 (a, b) – images positions are not equal, etc.) Please correct in all paper.
  2. Would be preferred that the article could be improved in case of results repeatability i.e. What was the counter-electrode material and shape? What was the volume of reactor? Were the voltage measured during process? Please provide any information available in the moment – maximum voltage, anything. The current is applied as square wave?
  3. What are the mechanical properties of the coating? Any characterization (scratch test, hardness) will improve the article quality.
  4. Comparation of coatings corrosion properties with unprocessed material would be also interesting.

In summary, presented results are very interesting however the editorial part must be improved and the language checked.

Autor’s provided very nicely prepared cross sections for SEM, what may have very different results for brittle and porous PEO coating. It would be very profitably for the scientific community if the authors present their methodology for pouring samples into resin and grinding them.

Reviewer 2 Report

The authors report on the Influence of Electrolyte Temperature on the Color Value of Black Plasma Electrolytic Oxidation Coatings on AZ31B Mg Alloy. They investigated the temperature influence on the properties of PEO coatings containing V species, especially coatings darkening. Although the manuscript is nicely written, some aspects are not enough well addressed.
1) Please add experimental details about the samples preparation. Did the authors always use fresh electrolyte for each sample? What was the volume of electrolyte used? How long takes to stabilize the temperature in the system? Did the authors consider that time needed for temperature stabilization influences the amount of NH3 evaporated?
2) Please justify in the introduction what is the advantage of V oxides introduction into PEO coating on Mg.
3) The authors wrote:
Abstract: “The higher the temperature was, the more NH3 escaped from the electrolyte and NH3 participating in the reaction decreased, resulting in a decrease in V2O3, increase in color value, and darker coating.” Darker or lighter?
Page 11, line 239: “The temperature influence on coating color value, therefore, was interpreted to be that, with increased temperature, NH3 escape increased in the electrolyte, leading to decreased NH3 in the reaction, such that the amounts of V2O3 increased and V2O5 decreased, yielding an RVO decrease.” Is it correct that the V2O3 increase? Should not be the opposite?
These aspects are also discussed in other places, so please be sure that conclusions are consistent also there. (Page 3, line 102 and Page 11, conclusion 1.)
4) Could you please explain the reason for using EDTA chelating agent, usually applied for complexing cationic species?
5) In the XRD section, I think, the authors should not suggest that V2O3 content decrease, because it is also true for all the other compounds (all the peaks decrease), since the coatings become thinner, and signals form Mg alloy substrate become more pronounced.
6) Do the authors sure that peak V2O5 is really present in the XRD spectrum? It looks like overlapped with Mg peak, and moreover one reflection from the compound is not enough to confirm its presence.
7) Fig. 2. Caption: The plural form of "spectrum" is “spectra”.
8) Please rewrite the sentence starting line 129, to clarify which compound forms which phase in the coating.
9) Table 6 – correct fonts size
10) In my opinion, XPS deconvolution should be improved (so also conclusions taken based on this section). Since p-type electrons give doublet signal it is required to show fitting for both spin-orbit peaks. Please be sure that fitting is done accordingly to principals e.g., constant FWHM value.
11) I would suggest not to focus only on one “maximum pore diameter” because this result can be very random and leads to wrong conclusions. Please make statistical analysis of pore-size distribution, or if not possible, discuss only general changes in coatings appearance.
12) XDR results discussed from line 177, I would recommend moving to the XRD section (page 4).
13) Fig 7. Please add a scale bar. Which coating is presented here?
14) References: 11 and 15 have reference numbers written twice.
I believe this contribution may be suitable for publication in Coatings journal after the above issues have been addressed.

Reviewer 3 Report

My comments are attached.

Round 2

Reviewer 2 Report

In my opinion, the authors did most of the corrections on a satisfactory level, however, the XPS deconvolution should be improved. It is not correct to show only half of the signal. In the case of p-type electrons, the peak has a doublet form and it is necessary to fit both spin-orbit peaks in the deconvolution process, keeping all the XPS deconvolution regimes. 

I believe that the work will be suitable for publishing after improving the XPS section. 
